# FedSecurity: A Benchmark for Attacks and Defenses in Federated Learning and Federated LLMs

## Abstract

This paper introduces FedSecurity, an end-to-end benchmark designed to simulate adversarial attacks and corresponding defense mechanisms in Federated Learning (FL). FedSecurity comprises two major components: FedAttacker, which simulates attacks injected during FL training, and FedDefender, which simulates defensive mechanisms to mitigate the impacts of the attacks. FedSecurity is open-source and can be customized to cover a wide range of machine learning models (*e.g.*, Logistic Regression, ResNet, and GAN) and federated optimizers (*e.g.*, FedAVG, FedOPT, and FedNOVA). We also demonstrate the use of FedSecurity during federated training of Large Language Models (LLMs), showcasing its adaptability and applicability in more complex scenarios.

## 1 Introduction

Federated Learning (FL) (McMahan et al., 2017a) facilitates training across distributed data and empowers individual clients to utilize their local data to collaboratively train machine learning models. Instead of sending their local data to a centralized server, FL clients train models on their local data and share the local models with the FL server, which aggregates the local models into a global model. This global model is redistributed to the clients, enabling the clients to further fine-tune the model using their local data.

FL maintains the privacy and security of client data by allowing clients to train locally without spreading their data to other parties. As a result of its privacy-preserving nature, FL has attracted considerable attention across various domains and has been utilized in numerous areas such as next-word prediction (Hard et al., 2018; Chen et al., 2019; Ramaswamy et al., 2019), hot-word detection (Leroy et al., 2019), financial risk assessment (Byrd & Polychroniadou, 2020), and cancer risk prediction (Chowdhury et al., 2022), demonstrating its wide-ranging versatility.

Recently, FL has found applications in large language models (LLMs) which expands its use cases. Referred to as *federated LLMs*, these models utilize FL during pre-training and finetuning as well as for prompt engineering (Chen et al., 2023). Currently, there are industry products that utilize FL (or distributed training) to train LLMs, including Deepspeed ZeRO (Rajbhandari et al., 2020; Wang et al., 2023), HuggingFace Accelerate (Gugger, 2021), Pytorch Lightning Fabric (Antiga, 2023). FL can facilitate LLM training due to the following reasons: *i) Distributed nature of LLM training data:* LLMs are pre-trained using large amounts of data, which often reside in different locations. Collecting such data to a central server is expensive and may also leak sensitive user information, while a viable way is to train LLMs in a federated manner. *ii) Scalability and efficiency:* LLMs, such as GPT-3 (Brown et al., 2020), have an extremely large number of parameters. Training LLMs on a single machine is infeasible and inflexible, while FL can be a good choice. *iii) Continuous improvement with user data:* LLMs can be deployed in a federated manner and local instances of the models can be further finetuned based on the local data, enabling the global model to improve over time based on users' data without ever having direct access to that data. This is particularly relevant for privacy-sensitive fields such as healthcare or personal communications.

Even though FL does not require sharing raw data with others, its decentralized and collaborative nature might inadvertently introduce privacy and security vulnerabilities. In recent years, a burgeoning body of research has spotlighted various attack mechanisms in FL (Bhagoji et al., 2019;

Xie et al., 2019; Lam et al., 2021; Jin et al., 2021; Tomsett et al., 2019; Chen et al., 2017; Fang et al., 2020; Tolpegin et al., 2020; Zhu et al., 2019; Bagdasaryan et al., 2020; Zhang et al., 2022a; Kariyappa et al., 2022; Zhang et al., 2022b), where adversarial clients might submit spurious models to disrupt the global model from converging, or sabotage the global model to misidentify particular data samples by planting backdoors. Meanwhile, a wide range of defense mechanisms has emerged to mitigate the impact of these attacks (Li et al., 2022; Kumari et al., 2023; Sun et al., 2019; Ozdayi et al., 2021; Blanchard et al., 2017; Xie et al., 2020; Chen et al., 2017; Sun et al., 2019; Karimireddy et al., 2020; Yin et al., 2018; Pillutla et al., 2022; Fung et al., 2020; Xie et al., 2021; Yin et al., 2018; Ma et al., 2022; Kumar et al., 2022; Chen et al., 2022). Despite the efforts for addressing the vulnerability of FL systems, there still lacks a comprehensive benchmark for comparing approaches under unified sittings. Moreover, existing research has not yet investigated applying the attack and defense mechanisms to federated LLMs. In contrast to traditional small models, LLMs are distinguished by the large number of parameters and complex training datasets obtained from unregulated sources, which could introduce challenges when applying attacks and defenses on top of them. These motivate a need for a standardized and comprehensive benchmark to assess baseline attack and defense mechanisms in the context of FL and federated LLMs.

To this end, this paper introduces FedSecurity, a benchmark that simulates attacks and defenses in FL.[1] FedSecurity comprises two primary components: FedAttacker and FedDefender. FedAttacker simulates attacks in FL to help understand and prepare for potential security risks, while FedDefender is equipped with various defense mechanisms to counteract the threats injected by FedAttacker. Besides small model tasks, we also apply FedSecurity to federated LLMs. Our contributions are summarized as follows:

*i*) **Enabling benchmarking of various attacks and defenses in FL**. FedSecurity implements attacks that are widely considered in the literature, including Byzantine attacks of random/zero/flipping modes (Chen et al., 2017; Fang et al., 2020), label flipping backdoor attack (Tolpegin et al., 2020), deep leakage gradient (Zhu et al., 2019), and model replacement backdoor attack (Bagdasaryan et al., 2020). Some of the well-known defense mechanisms supported include Norm Clipping (Sun et al., 2019), Robust Learning Rate (Ozdayi et al., 2021), Krum (and $m$-Krum) (Blanchard et al., 2017), SLSGD (Xie et al., 2020), geometric median (Chen et al., 2017), weak DP (Sun et al., 2019), CClip (Karimireddy et al., 2020), coordinate-wise median (Yin et al., 2018), RFA (Pillutla et al., 2022), Foolsgold (Fung et al., 2020), CRFL (Xie et al., 2021), and coordinate-wise trimmed mean (Yin et al., 2018).

*ii*) **Flexible configuration.** FedSecurity supports configurations using a .yaml file. Users can utilize two parameters, "enable_attack" and "enable_defense", to activate FedAttacker and FedDefender. Sample configurations are respectively shown in Figures 14 and Figures 15of Appendix A.

*iii*) **Supporting customization of attack and defense mechanisms.** We provide APIs in FedSecurity to enable users to integrate user-defined attacks and defenses in addition to the default baseline attack and defense mechanisms included in FedSecurity.

*iv*) **Supporting various models and FL optimizers.** FedSecurity can be utilized with a wide range of models, including Logistic Regression, LeNet (LeCun et al., 1998), ResNet (He et al., 2015), CNN (LeCun et al., 1989), RNN (Rumelhart et al., 1986), GAN (Goodfellow et al., 2014), and so on. FedSecurity is compatible with various FL optimizers, such as FedAVG (McMahan et al., 2016), FedSGD (Shokri & Shmatikov, 2015), FedOPT (Reddi et al., 2021), FedPROX (Li et al., 2020), FedGKT (He et al., 2020), FedGAN (Rasouli et al., 2020), FedNAS (He et al., 2021), Fed-NOVA (Wang et al., 2020b), and so on.

*v*) **Extensions to federated LLMs and real-world applications.** FedSecurity is suitable for demonstrating attacks and defenses during training of federated LLMs (Section 5.2). We also include a real-world experiment, where we use edge devices for FL with FedSecurity instead of simulations (Appendix E). These show the adaptability of the proposed FedSecurity benchmark.

*Key takeaways*: *i*) Byzantine attack of random mode (Chen et al., 2017; Fang et al., 2020) is effective in decreasing the test accuracy of the global model, and $m$-Krum (Blanchard et al., 2017) can produce robust results against various attacks; *ii*) while introducing a defense mechanism can help mitigate attacks, it might also affect the aggregation results, potentially compromising the model's

---

[1]FedSecurity library: [We do not release the implementation in order not to hurt our anonymity.]

performance. However, in actual FL systems, attacks are infrequent. Therefore, it's crucial to weigh the benefits against potential drawbacks before integrating a defense mechanism into real systems.

## 2 PRELIMINARIES AND OVERVIEW

In this section, we first discuss the related literature and introduce adversarial models considered in FedSecurity. Then we present an overview of FedSecurity.

### 2.1 RELATED WORKS

Recent years, various benchmarks have been introduced for FL, such as TensorFlow Federated (Abadi et al., 2015), PySyft (Ziller et al., 2021), FATE (Liu et al., 2021), Flower (Beutel et al., 2020), FedScale (Lai et al., 2022), NVIDIA FLARE (Roth et al., 2022), OpenFL (Reina et al., 2021), Fed-BioMed (Silva et al., 2020), IBM Federated Learning (Ludwig et al., 2020), Federated-Scope (Xie et al., 2022), and FLUTE (Dimitriadis et al., 2022). Among these, only FederatedScope delves into the implications of adversarial attacks in FL, with a focus on data reconstruction attacks that utilize models or gradients to revert sensitive information, including GAN-based leakage attack (Hitaj et al., 2017), Passive Property Inference (Melis et al., 2019), and DLG attack (Zhu et al., 2019). However, FederatedScope neglects to address attacks prevalent in the research literature, *e.g.*, Byzantine attacks (Yin et al., 2018; Yang et al., Dec 2019). It also does not include any defense mechanisms for FL. It is worth noting that, while FederatedScope integrates secret-sharing (Beimel, 2011), it is in the scope of federated analytics (Elkordy et al., 2023; Ramage, 2020; Wang et al., 2022a; Jung et al., 2012), instead of FL.

FedSecurity implements attacks that are widely considered in the literature (Yin et al., 2018; Tolpegin et al., 2020; Zhu et al., 2019); it also integrates a wide range of defense mechanisms (Sun et al., 2019; Ozdayi et al., 2021; Blanchard et al., 2017; Xie et al., 2020; Chen et al., 2017; Sun et al., 2019; Karimireddy et al., 2020; Yin et al., 2018; Pillutla et al., 2022; Fung et al., 2020; Xie et al., 2021; Yin et al., 2018). Designed with flexibility in mind, FedSecurity offers configurable settings and APIs, enabling users to customize their attack and defense mechanisms.

### 2.2 ADVERSARIAL MODEL

Real-world adversaries in FL systems fall into two categories: active and passive adversaries.

*Active Adversaries.* Active adversaries intentionally manipulate training data or trained models to achieve malicious goals. This might involve altering models to prevent global model convergence (*e.g.*, Byzantine attacks (Chen et al., 2017; Fang et al., 2020)), or subtly misclassifying a specific set of samples to minimally impact the overall performance of the global model (*e.g.*, backdoor attacks (Bagdasaryan et al., 2020; Wang et al., 2020a; Zhang et al., 2022a)). Active adversaries can take various forms, including: 1) malicious clients who manipulate their local models (Bagdasaryan et al., 2020; Chen et al., 2017; Fang et al., 2020; Zhang et al., 2022a) or submit contrived models without actual training (Wang, 2022); 2) a global "sybil" (Tolpegin et al., 2020; Fung et al., 2020) that has full access to the FL system and possesses complete knowledge of the entire system, including local and global models for each training round and clients' local datasets. This "sybil" may also modify data within the FL system, such as clients' local datasets and their submitted local models; and 3) external adversaries capable of monitoring the communication channel between clients and the server, thereby intercepting and altering local models during the transfer process.

*Passive Adversaries.* Passive adversaries do not modify data or models, but may still pose a threat to data privacy by potentially deducing sensitive information (such as local training data) from revealed models (gradients, or model updates) (Zhu et al., 2019). Examples of passive adversaries include: 1) an adversarial FL server attempting to infer local training data using submitted local models; 2) adversarial FL clients trying to deduce other clients' training data using the global model provided by the server; and 3) external adversaries, *e.g.*, hackers, that access communication channels to acquire local and global models transferred between clients and the FL server.

The adversaries can inject attacks at different stages of FL training. In summary, active adversaries can conduct attacks that modify local models (*model poisoning attacks*) or poison local datasets (*data poisoning attack*), while passive adversaries can infer sensitive information, such as user data,

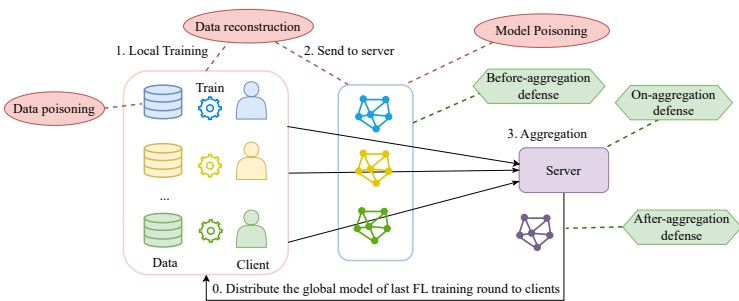

**Figure 1:** FedSecurity overview. FedSecurity enables injecting attacks (shown in red) and defenses (shown in green) at various stages of FL training at the clients and at the server.

based on the models or gradients they observe (*data reconstruction attacks*). In the next subsection, we illustrate how to inject those attacks at different stages of FL frameworks.

## 2.3 OVERVIEW OF FEDSECURITY

FedSecurity serves as an external component that injects attacks and defense mechanisms at different stages of training without altering the existing processes in FL. FedSecurity utilizes FedAttacker and FedDefender to initiate two instances and simulate attacks and defenses, respectively. The two instances are initialized once and are accessible by other objects in the FL system[2].

*Injection of attacks.* Without loss of generality, we classify the attacks in FL into the following three categories based on the targets of the attacks:

*i) Data poisoning attacks* that are conducted by active adversaries to modify clients' local datasets and are injected at clients (Tolpegin et al., 2020; Dang et al., 2021).

*ii) Model poisoning attacks* that are also conducted by active adversaries to temper with local models submitted by clients (Fang et al., 2020; Shejwalkar & Houmansadr, 2021; Bhagoji et al., 2019). FedAttacker injects these attacks before the aggregation of local models in each FL training round at the server, so that it can get access to all client models submitted in that training round.

*iii) Data reconstruction attacks* that are conducted by passive adversaries by exploring local models or updates to infer information about the training data (Melis et al., 2018; Zhang et al., 2020; Luo et al., 2021; Wang et al., 2022b; Fowl et al., 2021). FedAttacker injects such attacks at the FL server, as the FL server has access to all local models and the global model of each iteration, and can perform the attacks with flexibility.

*Injection of defenses.* FedDefender integrates defenses to mitigate, if not completely nullify, the impacts of the injected attacks. Since the defenses either address issues related to tampered local models by active adversaries[3] or prevent adversaries from deducing information from the local/global models shared between clients and the FL server, FedDefender deploys defenses at the FL server to get access to all local models and global models in each FL training round. For this, FedDefender can inject three functions at different stages of FL aggregation:

*i) Before-aggregation functions* that modify local models submitted by clients.

*ii) On-aggregation functions* that modify the FL aggregation function to mitigate the impacts of local models submitted by adversarial clients.

*iii) After-aggregation functions* that modify the aggregated global model (*e.g.*, by adding noise or clipping) to protect the real global model or improve its quality.

Figure 1 summarizes the injections of attacks and defenses to the FL framework in FedSecurity. We also provide detailed algorithms for injecting attacks and defenses to different stages of FL training, as shown in Algorithm 1 (for server aggregation) and Algorithm 2 (for client training) in Appendix B. Below, we explain the implementations of attacks and defenses in detail.

---

[2]Such design is achieved by the singleton design pattern (Gamma et al., 1995).

[3]Note that poisoning local datasets also results in tampered local models.

## 3    IMPLEMENTATION OF ATTACKS IN FEDATTACKER

FedAttacker injects model poisoning, data poisoning, and data reconstruction attacks at different stages of FL training and provides APIs for these attacks. We present each class of attacks and defer the user integration of a new attack to FedSecurity to Appendix C.1 due to space limitations.

### 3.1    MODEL POISONING ATTACKS

Model poisoning attacks are designed to modify the local models submitted by clients. FedAttacker injects such attacks before FL aggregation in each iteration, modifying each local model directly. Model poisoning attacks implemented in FedAttacker include Byzantine attacks (Chen et al., 2017; Fang et al., 2020) of three different modes and the model replacement backdoor attack (Bagdasaryan et al., 2020). For example, FedAttacker implements three modes of Byzantine attacks, as follows:

- *Zero mode* that poisons the client models by setting their weights to zero.
- *Random mode* that manipulates client models by attributing random values to model weights.
- *Flipping mode* that updates the global model in the opposite direction by formulating a poisoned local model based on the global model $\mathbf{w}_g$ and the real local model $\mathbf{w}_\ell$ as $\mathbf{w}_g + (\mathbf{w}_g - \mathbf{w}_\ell)$.

**APIs for Model Poisoning Attacks.**    FedAttacker has two APIs for model poisoning attacks.

- $poison\_model(local\_models, auxiliary\_info)$, which takes the local models submitted by clients in the current FL iteration and modifies the local models. The input $local\_models$ is a list of tuples containing the number of data samples and the submitted client models. The input $auxiliary\_info$ is any information used in the defense, *e.g.*, the global model in the last FL iteration.
- $is\_model\_poisoning\_attack()$, which checks whether the attack component is activated and whether the attack modifies local models.

### 3.2    DATA POISONING ATTACKS

Data poisoning attacks modify (or poison) local datasets of some clients to achieve some malicious goals, *e.g.*, degrading the performance of the global model or inducing the global model to misclassify some samples. As an example, in label flipping attack (Tolpegin et al., 2020), a global "sybil" controls some clients and modifies their local data by mislabeling samples of some classes to wrong classes. Given a source class (or label) $c_s$ and a target class $c_t$, the local dataset of each poisoned client is modified such that all samples with class $c_s$ are now associated with an incorrect label $c_t$.

**APIs for Data Poisoning Attacks.** FedAttacker has two APIs for data poisoning attacks.

- $poison\_data(dataset)$, which takes a local dataset and mislabels a set of chosen samples based on the clients' (or attackers') requirements, which are included in the configuration. Normally, clients would change labels of a specific subset of samples to some other labels in the same dataset, or label a set of samples to new classes that do not exist in the dataset.
- $is\_data\_poisoning\_attack()$, which examines whether FedAttacker is enabled and whether the attack requires poisoning the datasets.

### 3.3    DATA RECONSTRUCTION ATTACKS

Data reconstruction attacks are performed by passive adversaries that attempts to infer sensitive information without actively interfering with the FL training or the local data. We assume that there is no leakage during the local training process in FL, as clients are on their fully trusted local machines. Thus, data reconstruction attacks take the trained models (either the global model or the local models) to revert training data. For example, Deep Leakage from Gradients (DLG) attack (Zhu et al., 2019) infers local training data from the publicly shared gradients. A passive adversary can use the global model from the previous FL training round and the newly obtained model to compute a "model update" between models in different FL training rounds to deduce the training data.

**APIs for Data Reconstruction Attacks.** We have two APIs for data reconstruction attacks.

- $reconstruct\_data(model, auxiliary\_info)$, which takes a client model or a global model to reconstruct the training data. It also takes some extra information ($auxiliary\_info$) to help infer.

- $is\_data\_reconstruction\_attack()$, which examines whether the attack component is enabled and whether the attack requires reconstructing training data using the trained models.

# 4 IMPLEMENTATION OF DEFENSES IN FEDDEFENDER

FedDefender injects defense functions at different stages of FL aggregation at the server. Based on the point of injection, FedDefender provides three types of functions to support defense mechanisms, including 1) before-aggregation, 2) on-aggregation, and 3) after-aggregation. Note that a defense may inject functions at one or multiple stages of FL aggregation.

## 4.1 BEFORE-AGGREGATION DEFENSES

Before-aggregation functions operate on local models of each FL training iteration to mitigate (or eliminate) the impacts of potential attacks. We use Krum (Blanchard et al., 2017) as an example.

**Krum.** Krum (Blanchard et al., 2017) tolerates $f$ Byzantine clients among $n$ clients by retaining only one local model that is the most likely to be benign as the global model. That is, Krum selects a single model as the global model in aggregation. A generalization of Krum is $m$-Krum (Blanchard et al., 2017) that selects $m$ client models with the $m$ lowest scores for aggregation, instead of choosing only one local model. This approach requires less than $\frac{n-m}{2} - 1$ clients to be malicious.

**APIs for before-aggregation functions.** We provide two APIs for before-aggregation functions:

- $defend\_before\_aggregation(local\_models, auxiliary\_info)$, which modifies the client models of the current FL iteration. The input *local_models* is a list of tuples that contain the number of samples and the local model submitted by each client in the current FL iteration. The input *auxiliary_info* can be any information that is utilized in the defense functions.
- $is\_defense\_before\_aggregation()$, which checks whether the FedDefender is activated and whether the defense requires injecting functions before aggregating local models at the server.

## 4.2 ON-AGGREGATION DEFENSES

On-aggregation defense functions modify the aggregation function to a robust version that tolerates or mitigates impacts of the potential adversarial client models. As an example, RFA (Robust Federated Aggregation) (Pillutla et al., 2022) computes a geometric median of the client models in each iteration as the aggregated model, instead of simply averaging the client models. RFA defense effectively mitigates the impact of poisoned client models, as the geometric median can represent the central tendency of the client models, and the median point is chosen in a way to minimize the sum of distances between that point and the other client models of the current FL iteration. In practice, the geometric median is calculated using the Smoothed Weiszfeld Algorithm (Pillutla et al., 2022).

**APIs for on-aggregation defenses.** We provide two APIs for on-aggregation defense functions:

- $defend\_on\_aggregation(local\_models, auxiliary\_info)$, which takes the local models of the current training round for aggregation. The input *local_models* is a list of tuples that contain the number of samples and the local model submitted by each client in the current FL iteration. The input *auxiliary_info* can include any information required by the defense functions.
- $is\_defense\_on\_aggregation()$, which checks if the defense component is enabled and whether the current defense requires the injection of functions during aggregation.

## 4.3 AFTER-AGGREGATION DEFENSE

After-aggregation defense functions modify the aggregation result, *i.e.*, the global model, of each FL iteration to mitigate the effects of poisoned local models or protect the global model from potential adversaries. As an example, CRFL (Xie et al., 2021) clips the global model to bound the norm of the model each time after aggregation at the FL server. The FL server then adds Gaussian noise to the clipped global model before distributing the global model to the clients for the next FL iteration.

**APIs for After-Aggregation Defenses.** We provide two APIs to support after-aggregation defenses:

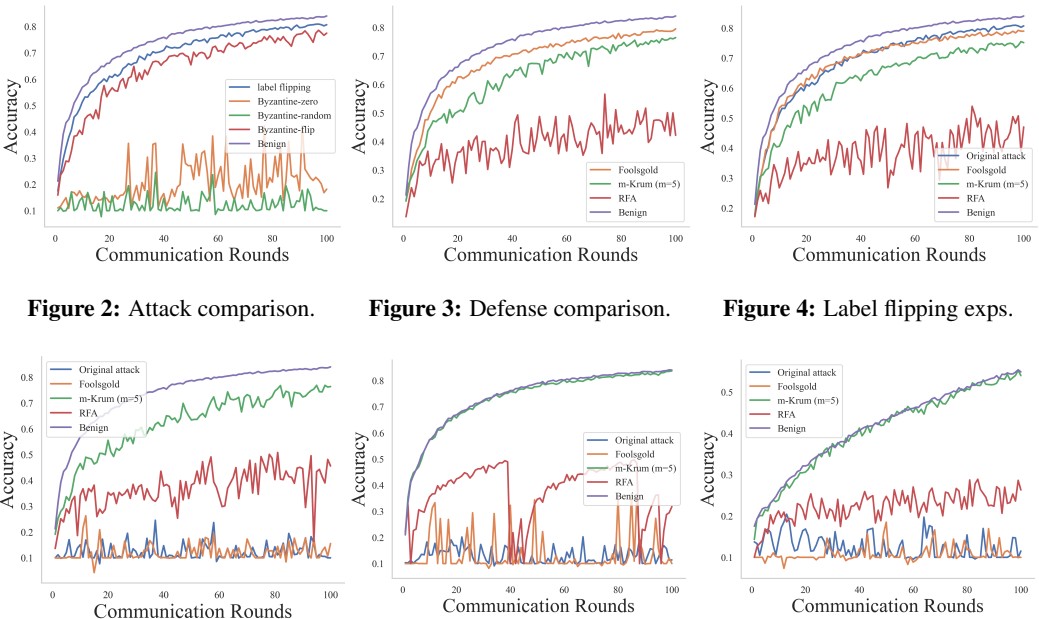

**Figure 2:** Attack comparison.     **Figure 3:** Defense comparison.     **Figure 4:** Label flipping exps.

**Figure 5:** Random-Byzantine exps.    **Figure 6:** I.I.D. data evaluations.    **Figure 7:** Scale # clients to 100.

- $defend\_after\_aggregation(global\_model)$, which directly modifies the global model after aggregation using methods such as clipping or adding noise.
- $is\_defense\_after\_aggregation()$, which checks if the defense component is activated and whether the current defense requires injecting functions after aggregation.

## 5 EXPERIMENTAL EVALUATIONS

This section presents a comprehensive evaluation of FedSecurity to benchmark some of the well-known attack and defense mechanisms in FL.

**Experimental setting.** A summary of datasets and models for evaluations can be found in Table 1 in Appendix D. By default, we employ ResNet20 and the non-i.i.d. CIFAR10 dataset (partition parameter $\alpha = 0.5$), as the non-i.i.d. setting closely captures real-world scenarios. We further extend our evaluations to i.i.d. cases and various other models and datasets. For evaluations on LLMs, we utilize FedLLM (FedML Inc., 2023) that trains LLMs in a federated manner. We employ the Pythia-1B model (Biderman et al., 2023) and PubMedQA (Jin et al., 2019), a non-i.i.d. biomedical research dataset that contains 212,269 questions for question answering. We utilize the "artificial" subset for training and the "labelled" subset for testing. We utilize FedAVG in our experiments. Evaluations are conducted on a server with 8 NVIDIA A100-SXM4-80GB GPUs.

### 5.1 EVALUATIONS ON FL

Unless otherwise noted, we use 10 clients, set the percentage of malicious clients to 10%, and evaluate results with the accuracy of the global model. We employ three attack mechanisms, including label flipping attacks and Byzantine attacks of random mode and flipping mode. For the label flipping attack, we set the attack to modify the local and test data labels of malicious clients from label 3 to label 9 and label 2 to label 1. We utilize three defense mechanisms: $m$-Krum (Blanchard et al., 2017), Foolsgold (Fung et al., 2020), and RFA (Pillutla et al., 2022). For $m$-Krum, we set $m$ to 5, which means 5 out of 10 submitted local models participate in aggregation in each training round.

**Exp 1: Attack Comparisons.** This experiment evaluates the impact of various attacks on test accuracy, using a no-attack scenario as a baseline. As illustrated in Figure 2, Byzantine attacks, specifically in the random and zero modes, substantially degrade accuracy. In contrast, the label flipping attack and the flipping mode of the Byzantine attack show a milder impact on accuracy. This can be attributed to the nature of Byzantine attacks, where Byzantine attackers would prevent

the global model from converging, especially for the random mode that generates weights for models arbitrarily, causing the most significant deviation from the benign local model. In subsequent experiments, unless specified otherwise, we employ the Byzantine attack in the random mode as the default attack, as it provides the strongest impact compared with the other three attacks.

**Exp 2: Defense Comparisons.** This experiment investigates the potential impact of defense mechanisms on accuracy in the absence of attacks, *i.e.*, whether defense mechanisms inadvertently degrade accuracy when all clients are benign. We incorporate a scenario without any defense or attack as our baseline. As illustrated in Figure 3, it becomes evident that when all clients are benign, involving defense strategies to FL training might lead to a reduction in accuracy. This decrease might arise from several factors: the exclusion of some benign local models from aggregation, *e.g.*, as in $m$-Krum, adjustments to the aggregation function, *e.g.*, as in RFA, or re-weighting local models, *e.g.*, as in Foolsgold. Specifically, the RFA defense mechanism significantly impacts accuracy as it computes a geometric median of the local models instead of leveraging the original FedAVG optimizer, which introduces a degradation in accuracy.

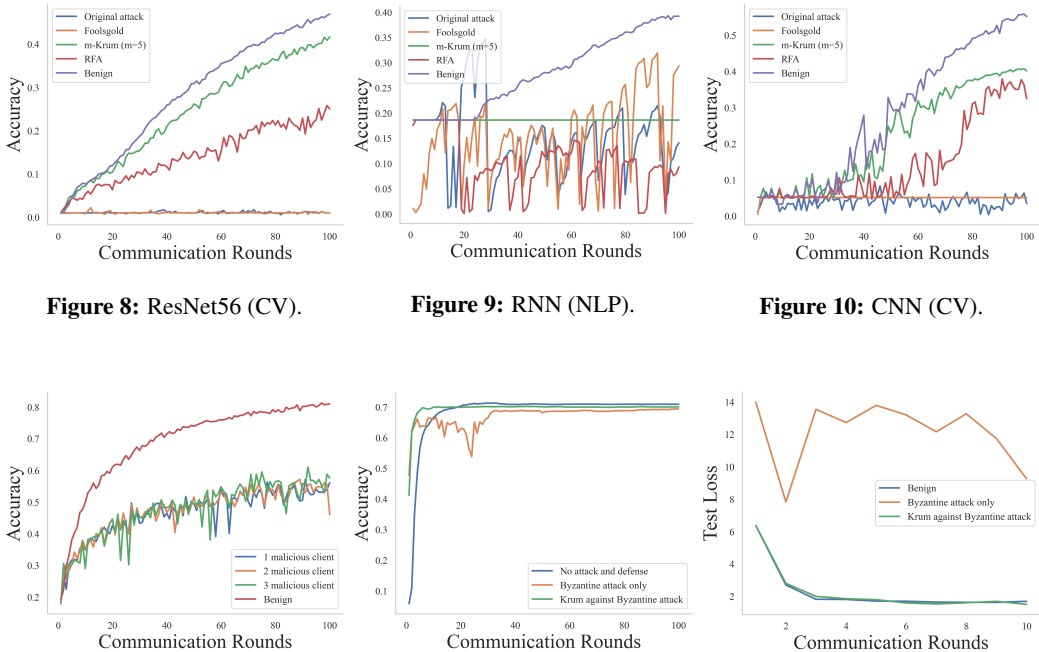

**Figure 8:** ResNet56 (CV).  **Figure 9:** RNN (NLP).  **Figure 10:** CNN (CV).

**Figure 11:** Varying # adversaries.  **Figure 12:** BERT evaluations.  **Figure 13:** Pythia-1B evaluations.

**Exp 3: Evaluations of defense mechanisms against activated attacks.** This experiment evaluates the effect of defense mechanisms in the context of ongoing attacks. We include two baseline scenarios: 1) an "original attack" scenario with an activated attack without any defense in place, and 2) a "benign" scenario with no activated attack or defense. We select label flipping attack and the random mode of Byzantine attack based on their impacts in **Exp1**, where label flipping has the least impact and the random mode of Byzantine attack exhibits the largest impact, as shown in Figure 2. Results for the label flipping and the random mode of Byzantine attacks are in Figure 4 and Figure 5, respectively. These results indicate that the defenses may contribute to minor improvements in accuracy for low-impact attacks, *e.g.*, Foolsgold in Figure 4. In certain cases, it is noteworthy that the defensive mechanisms may inadvertently compromise accuracy, such as the case with RFA in Figure 4. For high-impact attacks, such as the Byzantine attack of the random mode, Krum exhibits resilience, effectively neutralizing the negative impact of the attacks, as shown in Figure 5.

**Exp 4: Evaluations on i.i.d. data.** This experiment evaluates various defense mechanisms against an attack on i.i.d. data. We select the random mode of the Byzantine attack, and employ Foolsgold, $m$-Krum ($m = 5$), and RFA to counteract the adverse effects of this attack. As shown in Figure 6, $m$-Krum is the most effective one among all the defense mechanisms, where the test accuracy is close to the case where all the FL clients are honest, *i.e.*, no attack scenario.

**Exp 5: Scaling the number of clients to 100.** This experiment scales the number of clients to 100 and evaluates the defense mechanisms against the random mode of the Byzantine attack. We employ Foolsgold, $m$-Krum (with $m = 5$), and RFA to counteract the adverse effects of this attack. As shown in Figure 7, $m$-Krum is the most effective one among all the defense mechanisms, and the test accuracy is very close to the case where no attack happens.

**Exp 6: Evaluations on different models.** We evaluate defense mechanisms against the random mode of the Byzantine attack with different models and datasets, including: *i*) ResNet56 + CI-FAR100, *ii*) RNN + Shakespeare, and *iii*) CNN + FEMNIST. The results are shown in Figures 8, 9, and 10, respectively. The results show that while the defense mechanisms can mitigate the impact of attacks in most cases, some attacks may fail some tasks, *e.g.*, $m$-Krum fails RNN in Figure 9, and Foolsgold fails CNN in Figure 10. This is because the two defense mechanisms either select several local models for aggregation in each FL training round, or significantly re-weight the local models, which may eliminate some local models that are important to the aggregation in the first several FL training iterations, leading to unchanged test accuracy in later FL iterations.

**Exp 7: Varying the number of malicious clients.** This experiment evaluates the impact of varying numbers of malicious clients on test accuracy. We utilize $m$-Krum to protect against 1, 2, and 3 malicious clients out of 10 clients in each FL training round. As shown in Figure 11, the test accuracy remains relatively consistent across different numbers of malicious clients, as in each FL training round, $m$-Krum selects a local model that is the most likely to be benign to represent the other models, effectively minimizing the impact of malicious client models on the aggregation.

We present an experiment that utilizes real-world edge devices in Theta network (Theta Network., 2023) to showcase the scalability of FedSecurity to real-world applications in Appendix E.

## 5.2 EVALUATIONS ON FEDERATED LLMS

We employ two LLMs, BERT (Devlin et al., 2018) and Pythia (Biderman et al., 2023), to showcase the scalability of FedSecurity and its applicability to federated LLM scenarios. We notice that some defenses (*e.g.*, Foolsgold (Fung et al., 2020)) that require memorizing intermediate results, such as models of previous FL training rounds, might encounter limitations when integrated with LLMs due to the significant cache introduced. Considering this, we utilize $m$-Krum for our experiments, as it does not require storing intermediate results and demonstrates consistent performance in most of our previous experiments.

**Exp 8: Evaluations of Krum against model replacement backdoor attack on BERT.** This experiment utilizes BERT (Devlin et al., 2018) and the 20 news dataset (Lang, 1995) for a classification task. We employ 10 clients and set 1 client to be malicious in each FL training round. We set $m$ to 5 in $m$-Krum, *i.e.*, 5 out of 10 local models participate in aggregation in each FL training round. Results in Figure 12 show that $m$-Krum effectively mitigates the adversarial effect, bringing the accuracy closer to the level of the attack-free case.

**Exp 9: Evaluations of Krum against the Byzantine attack on Pythia-1B.** We employ 7 clients for FL training, and 1 out of 7 clients is malicious in each round of FL training. We set the $m$ parameter in $m$-Krum to 2, signifying that 2 out of 7 submitted local models participate in the aggregation in each FL training round. The performance is evaluated based on the test loss. Results in Figure 13 show that Byzantine attack significantly increases the test loss during training. Nevertheless, $m$-Krum effectively mitigates the adversarial effect.

## 6 CONCLUSION

This paper presents FedSecurity, a library designed to demonstrate potential adversarial attacks and corresponding defense strategies in FL to bolster innovation in the secure FL domain. FedSecurity contains two components: FedAttacker that simulates various attacks that can be injected during FL training, and FedDefender, which facilitates defense strategies to mitigate the impacts of these attacks. FedSecurity is open-sourced, and we welcome contributions from the research community to enrich the benchmark repository with novel attack and defense strategies to foster a diverse, comprehensive, and robust foundation for ongoing research in FL security.

## 7 ETHICS STATEMENT

FedSecurity is under the Apache 2.0 license, ensuring open access and customization. All datasets used for evaluations are publicly available, such as CIFAR10 (Krizhevsky et al., 2009), FEM-NIST (Caldas et al., 2018), Shakespeare (McMahan et al., 2017b), and so on. All models for evaluations are publicly available as well.

### 7.1 CODE OF ETHICS

**Data Handling and Protection.** We are aware of the risks associated with data processing in FL settings. Users can use the open-sourced FedSecurity library to simulate attacks and defenses on any machine without uploading their raw data and model. If users use our MLOps platform for simulation, only the model weights are uploaded. The uploaded model weights are encrypted (*i.e.*, only users with proper ownership can decrypt them) and can be deleted upon request. That is, we have no access to raw user data and we do not claim any data and model ownership.

**Benchmark Model Documentation and Transparency.** We are committed to: *i*) providing comprehensive documentation on the functionalities of the benchmark; *ii*) making a detailed datasheet available for the benchmark model, outlining its specifications, capabilities, and intended use cases; and *iii*) offering transparent and well-documented APIs for users.

### 7.2 LIMITATIONS AND FURTHER IMPROVEMENT

While FedSecurity offers a foundation for ML security research, we recognize its limitations and potential for further enhancement. Our plans for improvement are as follows: *i*) conducting more experiments on federated LLMs to provide a comprehensive understanding of vulnerabilities of LLMs within the FL context; and *ii*) designing and implementing advanced defense mechanisms against potential adversaries in asynchronous FL scenarios.

### 7.3 POTENTIAL NEGATIVE SOCIAL IMPACTS

Even though we put our best efforts in mitigating negative social impacts, the proposed FedSecurity benchmark might still be subject to some indistinct negative social impact, including:

- **Potential misuse**: While our module simulates attacks and defenses in FL to help the communities to better understand and compare the attacks in FL, it is not immune to malicious use. The platform could potentially be used to exploit vulnerabilities or develop advanced attack techniques in FL systems.
- **Data security**: FL is susceptible to various threats such as data poisoning. We acknowledge these inherent risks and are actively working on introducing defenses mechanisms to mitigate such attacks.
- **Privacy Concerns**: Although FL aims to train models without sharing raw data, there remains a risk of indirect data leakage, for example, attackers might utilize the models to infer whether specific data points are in the training datasets, where users should be cautious and informed.

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

APPENDIX

## A  EXAMPLE CONFIGURATION FILES FOR ATTACKS AND DEFENSES

We provide example configuration files for Byzantine attack (Chen et al., 2017; Fang et al., 2020) in Figure 14 and for $m$-Krum defense (Blanchard et al., 2017) in Figure 15.

```
attack_args:
  enable_attack: true
  attack_type: byzantine
  attack_mode: random
  byzantine_client_num: 1
```

**Figure 14:** Configuration for Byzantine attack (Chen et al., 2017; Fang et al., 2020).

```
defense_args:
  enable_defense: true
  defense_type: krum
  krum_param_m: 5
  byzantine_client_num: 1
```

**Figure 15:** Configuration for $m$-Krum (Blanchard et al., 2017) defense.

## B    Algorithms for FL Server Aggregation and Client Training

The algorithms for injecting attacks and defenses in FL training are described in Algorithm 1 (for FL server aggregation) and Algorithm 2 (for client training).

---

**Algorithm 1: Server Aggregation**

---

**Inputs:** $\mathbf{w}'_g$: the global model of last FL training round; $\mathcal{W}_l$: the list of local models submitted by each client in the current FL training round.

**Variables:** $\mathcal{A}$: A FedAttacker instance initialized based on the FL configuration file; $\mathcal{D}$: A FedDefender instance that is initialized based on the FL configuration file.

1 **Function** $server\_aggregation(\mathcal{W}_l)$ **begin**
2     $\mathcal{W}_l \leftarrow before\_aggregation\_process(\mathcal{W}_l, \mathbf{w}'_g)$
3     $\mathbf{w}_g \leftarrow before\_aggregation\_process(\mathcal{W}_l, \mathbf{w}'_g)$
4     **return** $after\_aggregation\_process(\mathcal{W}_l, \mathbf{w}_g)$

5 **Function** $before\_aggregation\_process(\mathcal{W}_l, \mathbf{w}'_g)$ **begin**
6     **if** $\mathcal{A}.is\_attack\_enabled()$ **then**
7        **if** $\mathcal{A}.is\_data\_reconstruction\_attack()$ **then** $\mathcal{A}.reconstruct\_data(\mathcal{W}_l, \mathbf{w}'_g)$ ;
       **if** $\mathcal{A}.is\_model\_poisoning\_attack()$ **then** $\mathcal{W}_l \leftarrow \mathcal{A}.poison\_model(\mathcal{W}_l, \mathbf{w}'_g)$;
8     **if** $\mathcal{D}.is\_defense\_enabled()$ & $\mathcal{D}.is\_defense\_before\_aggregation()$ **then**
       $\mathcal{W}_l \leftarrow \mathcal{D}.defend\_before\_aggregation(\mathcal{W}_l, \mathbf{w}'_g)$
9     **return** $\mathcal{W}_i$

10 **Function** $on\_aggregation\_process(\mathcal{W}_l, \mathbf{w}_g)$ **begin**
11     **if** $\mathcal{D}.is\_defense\_enabled()$ & $\mathcal{D}.is\_defense\_on\_aggregation()$ **then**
       **return** $\mathcal{D}.defend\_on\_aggregation(\mathcal{W}_l, \mathbf{w}_g)$
12     **return** $aggregate(\mathcal{W}_i)$

13 **Function** $after\_aggregation\_process(\mathbf{w}_g)$ **begin**
14     **if** $\mathcal{D}.is\_defense\_enabled()$ & $\mathcal{D}.is\_defense\_after\_aggregation()$ **then**
       **return** $\mathcal{D}.defend\_after\_aggregation(\mathbf{w}_g)$
15     **return** $\mathbf{w}_g$

---

**Algorithm 2: Client Training**

---

**Inputs:** $dataset$: the local dataset of a client.

**Variables:** $\mathcal{A}$: A FedAttacker instance initialized based on the FL configuration file;

1 **Function** $client\_training(dataset)$ **begin**
2     **if** $\mathcal{A}.is\_attack\_enabled()$ & $\mathcal{A}.is\_data\_poisoning\_attack()$ **then**
       $dataset \leftarrow \mathcal{A}.poison\_data(dataset)$
3     $\mathbf{w}_l \leftarrow train(dataset)$
4     $send\_to\_server(\mathbf{w}_l)$

---

## C    Integration of New Attacks and Defenses

### C.1    Integration of a New Attack

To customize a new attack, users should follow these steps: *i*) determine the type of the attack, *i.e.*, model poisoning, data poisoning, or data reconstruction; *ii*) create a new class for the attack and implement functions using the APIs, *e.g.*, $attack\_model(*)$, $poison\_data(*)$, and $reconstruct\_data(*)$, to inject attacks at the appropriate stages of FL training; and *iii*) add the attack name to the corresponding enabler functions, *i.e.*, $is\_model\_poisoning\_attack()$, $is\_data\_poisoning\_attack()$, and $is\_data\_reconstruction\_attack()$, within the FedAttacker class to ensure that the injected attacks are activated at the proper stages of FL training.

## C.2 Integration of a New Defense

To implement a self-designed defense mechanism, users should first determine the stages to inject the defense functions (*i.e.*, before/on/after-aggregation), add a class for the new defense and implement the corresponding defense functions using the aforementioned APIs, *i.e.*, $defend\_before\_aggregation(*)$, $defend\_on\_aggregation(*)$, and $defend\_after\_aggregation(*)$, to inject functions at appropriate stages of FL. Note that some defenses involve more than one stage; thus, users need to implement all relevant functions. Users should add the name of the defense to the enabler functions to activate the injected function at the different stages of FL. The approach computes some scores using local models submitted by clients, and uses the scores to identify outlier local models before aggregating the local models. As such process only happens before aggregation, we only need to implement $defend\_before\_aggregation(*)$ for the defense class, and include the name of the defense in $is\_defense\_after\_aggregation()$.

## D Models and datasets for evaluations

Models and datasets used in this work are given in Table 1.

| Model | Dataset |
|---|---|
| ResNet20 (He et al., 2016) | CIFAR10 (Krizhevsky et al., 2009) |
| ResNet56 (He et al., 2016) | CIFAR100 (Krizhevsky et al., 2009) |
| CNN (McMahan et al., 2017a) | FEMNIST (Caldas et al., 2018) |
| RNN (bi-LSTM) (McMahan et al., 2017a) | Shakespeare (McMahan et al., 2017b) |
| BERT (Devlin et al., 2018) | 20News (Lang, 1995) |
| Pythia-1B (Biderman et al., 2023) | PubMedQA (Luo et al., 2022) |

**Table 1:** Models and datasets for evaluations.

## E Supplementary Experiment

In this section, to demonstrate the scalability of our benchmark, we include an experiment using real-world devices, instead of simulations.

**Exp10: Evaluations in real-world applications.** We utilize edge devices from the Theta network (Theta Network., 2023) to validate the scalability of FedSecurity to real-world applications. The FL client package is integrated into Theta's edge nodes, which periodically fetches data from the Theta back-end. Subsequently, the FL training platform capitalizes on these Theta edge nodes and their associated data to train, fine-tune, and deploy machine learning models.

We select $m$-Krum as the defense and the Byzantine attack of random mode as the attack. Considering the challenges posed by real-world environments, such as devices equipped solely with CPUs (lacking GPUs), potential device connectivity issues, network latency, and limited storage on edge devices (for instance, some mobile devices might have less than 500MB of available storage), we choose a simple task by employing the MNIST dataset for a logistic regression task.

In our experimental setup, we deploy 70 client edge devices, designating 7 of these as malicious for each FL training round. For $m$-Krum, we set $m$ to 35, meaning that 35 out of the 70 local models are involved in aggregation during each FL training round. As illustrated in Figure 18, $m$-Krum mitigates the adversarial effect of the random-mode Byzantine attack. We also include a screenshot of the platform, as shown in Figure 16 for the FL training process and Figure 17 for the training status of each device.

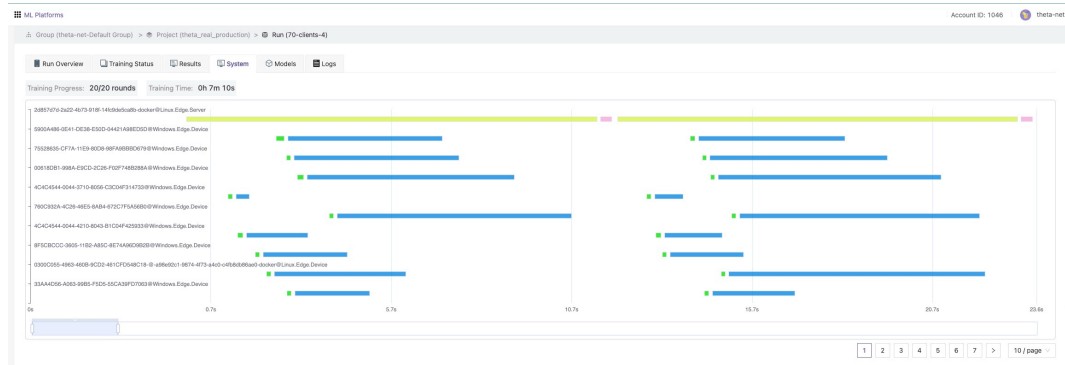

**Figure 16:** Real-world application. Yellow: aggregation server waiting time; pink: aggregation time; green: client training time; blue: client communication.

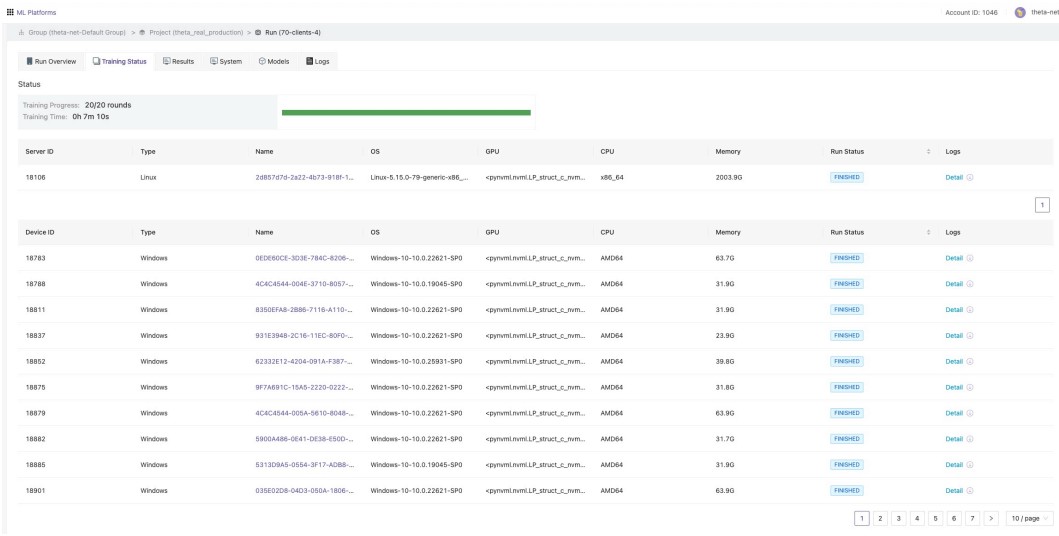

**Figure 17:** Real-world application: training status of devices.

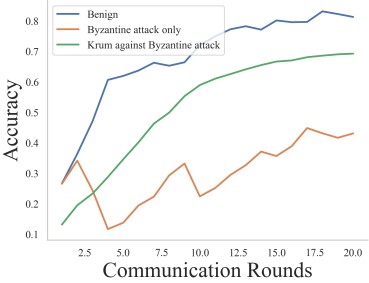

**Figure 18:** $m$-Krum against random-mode Byzantine attack in a real-world application.

