# OpenReview forum: "FedSecurity: A Benchmark for Attacks and Defenses in Federated Learning and Federated LLMs"
_ICLR.cc/2024/Conference — ICLR 2024 Conference Withdrawn Submission_

### Official Review · Reviewer_ozf6 · 2023-10-30

**Soundness:** 3 good
**Presentation:** 3 good
**Contribution:** 3 good
**Rating:** 6
**Confidence:** 4

**Summary:**

This paper presents a benchmark for federated learning security, including two components: federated attacker and federated defender. The federated attacker implements about eight classical attack methods covering data poisoning attacks, model poisoning attacks, and data reconstruction attacks. The defenders include before-aggregation, after-aggregation, and on-aggregation defenses.

**Strengths:**

A substantive assessment of the strengths of the paper, touching on each of the following dimensions: originality, quality, clarity, and significance. We encourage reviewers to be broad in their definitions of originality and significance. For example, originality may arise from a new definition or problem formulation, creative combinations of existing ideas, application to a new domain, or removing limitations from prior results. You can incorporate Markdown and Latex into your review.
S1. This paper proposes a comprehensive benchmark for federated learning security, which is expected to prompt the prosperity of security research of federated learning.
S2. The benchmark attempts to incorporate large language models, indicating its generalization ability.
S3. The limitations and future direction have been discussed in this paper.

**Weaknesses:**

W1. This paper highlights that the benchmark considers LLMs; however, the unique challenges/differences between using LLMs and classical models are unclear.

W2. It seems that the implemented attack/defense methods were published before 2021. The authors are encouraged to reproduce more SOTA attacks/defenses to improve the utility of this benchmark further.

W3. In footnote 2, the authors claim that their benchmark focuses on security rather than privacy. However it seems that “data reconstruction attack” is about privacy, which makes the taxology of this paper a little bit confusing.

**Questions:**

See Weeknesses. Addressing these weaknesses will further improve the convincing and quality of this paper.

---

> ### Author Response · Authors · 2023-11-21
>
> We appreciate your valuable feedback and your effort in reviewing our paper. Below are our responses.
>
> **W1: unique challenges/differences between using LLMs and classical models**
>
> We appreciate your valuable feedback. We have added the following description that mentions the differences between LLMs and traditional ML models in the introduction section.
> ```
> Moreover, existing research has not yet investigated applying the attack and defense mechanisms to federated LLMs. In contrast to traditional small models, LLMs are distinguished by the large number of parameters and complex training datasets obtained from unregulated sources, which could introduce challenges when applying attacks and defenses on top of them.
> ```
> Thus, investigating the performance of the most common and widely utilized federated learning attack and defense mechanisms in the context of federated LLMs is a valuable endeavor that may stimulate a deeper understanding of the security concept in federated LLMs.
>
> **W2: reproduce more SOTA attacks/defenses**
>
> Thanks for your feedback. So far, we have implemented 14 defenses and 8 attacks (see https://gitfront.io/r/Submission1595/4rUJj4M2NPzy/FedSecurity/blob/readme.md  for a full list). During our implementation, we selected the research papers based on their importance and citations to ensure the widely considered methods are implemented. As a result, the papers published in early years tend to be selected. Our benchmark is under active maintenance, and we definitely would like to make it more comprehensive in the future, as the research on security in FL progresses.
>
> **W3: confusing footnote about privacy benchmark**
>
> Thanks for your feedback! We acknowledge that the footnote was rather confusing, and we have removed it. Basically, our attack and defense library serves as a complementary part of an open-sourced FL benchmark, where we enable the attacks or defenses by activating the security components using a configuration file. Similarly, the FL benchmark includes a privacy library, i.e., a differential privacy (DP) library, as well, which can protect the models against the data construction attacks. To enable privacy-related attacks and defenses, we can activate attacks in the FedSecurity and a DP mechanism in the DP library.

---

### Official Review · Reviewer_5b7s · 2023-11-01

**Soundness:** 3 good
**Presentation:** 2 fair
**Contribution:** 3 good
**Rating:** 5
**Confidence:** 3

**Summary:**

The authors introduce a comprehensive benchmark for simulating adversarial attacks and their corresponding defense strategies in the context of Federated Learning (FL). This benchmark, known as FedSecurity, is composed of two primary components: FedAttacker, which replicates attacks introduced during FL training, and FedDefender, which emulates defense mechanisms aimed at mitigating the effects of these attacks. FedSecurity is an open-source tool that can be tailored to encompass a wide array of machine learning models (e.g., Logistic Regression, ResNet, and GAN) and federated optimization techniques (e.g., FedAVG, FedOPT, and FedNOVA). Additionally, the authors demonstrate the utility of FedSecurity in the context of federated training for Large Language Models (LLMs), showcasing its adaptability and relevance in more intricate scenarios.

**Strengths:**

1. It is important to have federated Large Language Models (LLMs) benchmark at this point for the community. I look forward the authors can dig deeper in this category and provide more complete/efficient implementation.

2. Different categories of attack and defense methods been implemented.

**Weaknesses:**

1. To implement federated LLM, the bottleneck is always the computational power. I wonder if the authors can provide some of their pre-trained models before/after fine-tunning?

2. Large-scale experiment up to 1000 clients is usually needed for benchmark works.

3. A table for the real training time is suggested (better on different devices).

4. Need more ablation studies given it is a benchmark work.

5. Attacks/defenses for NLP tasks (e.g., [1]) can be added to the benchmark for LLM and/or smaller models.

6. More backdoor attacks/defenses can be considered, and different evaluation metrics are needed (e.g., backdoor attack success rate)

[1] Zhang, Z., Panda, A., Song, L., Yang, Y., Mahoney, M., Mittal, P., Kannan, R. and Gonzalez, J., 2022, June. Neurotoxin: Durable backdoors in federated learning. In International Conference on Machine Learning (pp. 26429-26446). PMLR.

**Questions:**

Compared with existing federated learning benchmarks (e.g., Leaf [1], Flower [2]), what are the major advantages of the newly proposed benchmark?

[1] Caldas, S., Duddu, S.M.K., Wu, P., Li, T., Konečný, J., McMahan, H.B., Smith, V. and Talwalkar, A., 2018. Leaf: A benchmark for federated settings. arXiv preprint arXiv:1812.01097.
[2] Beutel, D.J., Topal, T., Mathur, A., Qiu, X., Fernandez-Marques, J., Gao, Y., Sani, L., Li, K.H., Parcollet, T., de Gusmão, P.P.B. and Lane, N.D., 2020. Flower: A friendly federated learning research framework. arXiv preprint arXiv:2007.14390.

---

> ### Author Response · Authors · 2023-11-21
> **Responses to Reviewer 5b7s**
>
> We greatly appreciate the reviewer's constructive feedback. Below are our responses.
>
> **W1: Provide pretrained models**
>
> Sure of course. In fact, we are planning to release our own LLMs. But due to the rule of anonymity we cannot discuss more about it here :)
>
> **W2: Large-scale experiments up to 1000 clients are usually needed for benchmark works.**
>
> We appreciate your feedback regarding the scale of the number of clients. First, we would like to point out that, to the best of our knowledge, so far, there are no real-world FL applications in the industry that utilize more than 100 clients. Typically, the real FL applications use fewer than 20 clients. We acknowledge the importance of large-scale experiments in FL benchmarks, and in Experiment 5, we deliberately chose to scale the number of devices to 100 based on a careful consideration of both feasibility and relevance to real-world scenarios. While the theoretical appeal of scaling up to 1000 clients is undeniable, the practical utility and relevance of such a scale are not yet established in real-world FL applications. Additionally, we included an experiment in real-world applications; see Exp10 in the appendix for details. In that experiment, we utilize 70 real edge devices from the Theta network, instead of just simulating, to validate the scalability of our benchmark to real-world applications.
>
> **W3: A table for the real training time is suggested (better on different devices).**
>
> Thanks for your suggestion! We included an experiment (Exp10 in Appendix) that evaluates our benchmark using 70 real-world devices from Theta Network. We also include the details for training on each device in Figure 16, where the local models are trained in an asynchronous way. From Figure 16 we can clearly see the training time for each device in each iteration. We also included detailed information for each device in Figure 17. The description of the experiment is as follows:
> ```
> Exp10: Evaluations in real-world applications. We utilize edge devices from the Theta network to validate the scalability of FedSecurity to real-world applications. The FL client package is integrated into Theta’s edge nodes, which periodically fetches data from the Theta back-end. Subsequently, the FL training platform capitalizes on these Theta edge nodes and their associated data to train, fine-tune, and deploy machine learning models. We select m-Krum as the defense and the Byzantine attack of random mode as the attack. Considering the challenges posed by real-world environments, such as devices equipped solely with CPUs (lacking GPUs), potential device connectivity issues, network latency, and limited storage on edge devices (for instance, some mobile devices might have less than 500MB of available storage), we choose a simple task by employing the MNIST dataset for a logistic regression task. In our experimental setup, we deploy 70 client edge devices, designating 7 of these as malicious for each FL training round. For m-Krum, we set m to 35, meaning that 35 out of the 70 local models are involved in aggregation during each FL training round. As illustrated in Figure 18, m-Krum mitigates the adversarial effect of the random-mode Byzantine attack. We also include a screenshot of the platform, as shown in Figure 16 for the FL training process and Figure 17 for the training status of each device.
> ```

---

> ### Author Response · Authors · 2023-11-21
>
> **W4: more ablation studies**
>
> In our paper, we have performed 10 different experiments to show how the most common and established attacks and defenses in FL can affect the model accuracy and loss. These include various CV and NLP experiments (with LLMs), as well as a real-world experiment that utilizes 70 edge devices from Theta Network. In the experiments, we visualize the performance under various configurations with varying number of clients, malicious clients, data distributions, and models. We also include key takeaways of this paper in the introduction section.
>
> ```
> Key takeaways: i) Byzantine attack of random mode is effective in decreasing the test accuracy of the global model, and m-Krum can produce robust results against various attacks; ii) while introducing a defense mechanism can help mitigate attacks, it might also affect the aggregation results, potentially compromising the model’s performance. However, in actual FL systems, attacks are infrequent. Therefore, it’s crucial to weigh the benefits against potential drawbacks before integrating a defense mechanism into real systems.
> ```
> If the reviewer has other suggestions, we would be happy to include them in our paper.
>
>
> **W5 & W6: implement more attacks and defenses**
>
> Thanks for your feedback! Currently, we have implemented 14 defenses and 8 attacks (see https://gitfront.io/r/Submission1595/4rUJj4M2NPzy/FedSecurity/blob/readme.md  for a full list). Our benchmark is under active maintenance, and we definitely would like to make it more comprehensive in the future. For the scope of this paper, our primary goal is to maintain a fair comparison of different attacks, including different modes of Byzantine attacks that aim to prevent the global model from converging, as well as backdoor attacks that try to utilize a malicious local model to substitute the global model (i.e., model replacement attack). In this case, the test accuracy is the best fit to evaluate the effect of different types of defense against the Byzantine and the backdoor attacks. We appreciate your valuable feedback, and we have cited the paper you suggested. We acknowledge backdoor success rate is an important metric to evaluate backdoor attacks. Considering the scope and the page limitation of this paper, we decided not to involve it for comparison in this paper. However, we might write an extension version of this benchmark paper in the future, where we focus more on different types of backdoor attacks, and we believe the backdoor success rate would be a good fit for evaluation.
>
>
> **Q1: Advantages compared to Leaf and Flower.**
>
> As far as we know, Leaf and Flower do not have a security module that compares the state-of-the-art attack and defense mechanisms in federated learning. Thus, the main contribution and advantage of our benchmark compared to the existing ones is the fact that we provide a comprehensive testing ground for the most common and widely utilized techniques in federated learning security.

---

> > ### Comment · Reviewer_5b7s · 2023-11-22
> >
> > I appreciate the authors' response and informative clarification. After reading the rebuttal carefully, I am inclined to maintain my original score. My decision is primarily based on the belief that the paper needs revisions to tackle issues related to variety and scalability. It would be beneficial for the paper to categorize and explain more attacks (including backdoor and Byzantine) and defenses (such as aggregation-based and post-training stage) across various categories, specifically in the context of image classification and/or NLP tasks, given that FL security is recognized as the main contribution.

---

### Official Review · Reviewer_r7rM · 2023-11-01

**Soundness:** 2 fair
**Presentation:** 3 good
**Contribution:** 2 fair
**Rating:** 5
**Confidence:** 4

**Summary:**

This paper introduces FedSecurity, a benchmark for simulating attacks and defenses in federated learning (FL).  FedSecurity has two main components: FedAttacker to inject attacks like data poisoning, model poisoning, and data reconstruction; and FedDefender to implement defenses like clipping, robust aggregation, and adding noise. It supports customizing attacks and defenses using provided APIs. It is also flexible to configure different models, datasets, and FL optimizers like FedAvg, FedOpt, etc. Experiments show Byzantine attacks like random noise can significantly degrade accuracy while defenses like m-Krum can mitigate it. Defenses may also inadvertently hurt accuracy when no attack happens. FedSecurity is also extended to federated training of large language models (LLMs) like BERT and Pythia. m-Krum defense is shown to be effective against backdoor and Byzantine attacks on LLMs.

**Strengths:**

1.	LLM extension: The benchmark is extended to federated training of large language models like BERT and Pythia, showing wider applicability.
2.	Real-world demonstration: A real-world experiment using edge devices shows the scalability beyond simulations.
3.	Analysis and insights: The experiments analyze impacts of attacks and defenses, highlighting the need to balance robustness vs potential negative impacts of defenses.

**Weaknesses:**

1.	Limited defense mechanisms: Only a small subset of defenses from the literature are implemented so far. More defenses could be included for completeness.
2.	Limited analysis: More in-depth analysis and visualization of how the attacks and defenses impact the model convergence would be useful.
3.	Small LLM experiments: Evaluations on large language models are limited to just BERT and Pythia. More experiments on diverse LLMs would strengthen this part.
4.	Narrow task types: Most experiments focus on image classification. Expanding the tasks to include NLP, recommendation systems, etc. would make it more representative.

**Questions:**

See in Weakness

---

> ### Author Response · Authors · 2023-11-21
> **Response to Reviewer r7rM**
>
> We sincerely appreciate your valuable feedback. Please find our detailed responses below.
>
> **W1: Limited defense mechanisms.**
>
> Thank you for your feedback regarding the scope of defense mechanisms. Currently, we have implemented 14 defense mechanisms that are widely recognized and considered in the literature. These mechanisms have been carefully chosen based on their prominence and relevance; see [1] for a full list of all the defense mechanisms we have implemented. Due to the page limit constraints of the submission, we made a deliberate choice to focus on several defenses that are the most commonly considered in the literature and are regarded as the state-of-the-art. This ensures that our paper remains focused to provide valuable insights into the most impactful and widely used defense mechanisms in the field. Our paper provides a thorough and detailed evaluation for the selected defense mechanisms, rather than a superficial overview of a broader range.
> [1] List of implemented attack and defense mechanisms: https://gitfront.io/r/Submission1595/4rUJj4M2NPzy/FedSecurity/blob/readme.md
>
> **W2: Limited analysis**
>
> In our paper, we have performed 10 different experiments to show how the most common and established attacks and defenses in FL can affect the model accuracy and loss. These include various CV and NLP experiments (with LLMs), as well as a real-world experiment that utilizes 70 edge devices from Theta Network. In the experiments, we visualize the performance under various configurations with varying number of total clients and malicious clients, as well as with different data distributions and models. We also include key takeaways of this paper in the introduction section.
>
> ```
> Key takeaways: i) Byzantine attack of random mode is effective in decreasing the test accuracy of the global model, and m-Krum can produce robust results against various attacks; ii) while introducing a defense mechanism can help mitigate attacks, it might also affect the aggregation results, potentially compromising the model’s performance. However, in actual FL systems, attacks are infrequent. Therefore, it’s crucial to weigh the benefits against potential drawbacks before integrating a defense mechanism into real systems.
> ```
> If the reviewer has other suggestions, we would be happy to include them in our paper.
>
>
> **W3: Small LLM experiments**
>
> Thanks for your feedback. Our initial decision to limit the evaluation to BERT and Pythia was guided by their prominence and representativeness in the field. This approach allowed us to establish a foundational understanding applicable to a broad spectrum of LLM applications. In response to your suggestion, we plan to include an experiment on Falcon-7B to provide a more comprehensive evaluation of the attack and defense mechanisms on LLMs.
>
> **W4: Narrow task types**
>
> Thank you for your feedback regarding the scope of tasks in our experiments. We would like to clarify that our study, in addition to focusing on image classification, indeed includes a wide range of tasks, as follows.
> NLP Experiments: We have incorporated NLP tasks in our paper, where we utilize two LLMs, BERT and Pythia, to test the attacks and defenses in a NLP context; see Exp 8 and Exp 9.
> Real-World Application with Logistic Regression: We also include an experiment that utilizes real-world edge devices in Theta network to showcase the scalability of FedSecurity to real-world applications. We utilize logistic regression and scale the number of clients to 70  to showcase the scalability of our library; see Exp10 in  Appendix E.

---

> > ### Comment · Reviewer_D1fe · 2023-11-22
> >
> > Thank you very much for addressing the points in my review. I really appreciate that the authors provided the code implementation and the catalogue with the different attacks and defenses. However, I agree with other reviewers that the experiments conducted can be improved and evaluation on scenarios with more clients to test the scalability is necessary. I increased my score to account for the authors' effort in addressing my comments, but I still think that the contribution of the paper is a bit limited.

---

### Official Review · Reviewer_D1fe · 2023-11-03

**Soundness:** 3 good
**Presentation:** 3 good
**Contribution:** 2 fair
**Rating:** 5
**Confidence:** 5

**Summary:**

The paper introduces FedSecurity, a library with attacks and defenses for federated learning useful for benchmarking and assessing the quality of defenses against different sets of attacks. The library comprises of two components: 1) FedAttack for training-time attacks in FL, including data and model poisoning and data reconstruction attacks; 2) FedDefender, which includes defenses that can be applied at the aggregator before, during or after the aggregation of the model parameters.

**Strengths:**

+ Robustness of Federated learning is a hot topic and libraries for assessing the robustness of different aggregation methods and defensive techniques systematically is useful not only for the research community, but also for other ML practitioners and developers.

+ The library seems to include a good set of attacks and defenses, although a table or a list with the complete catalogue of attacks and defenses would be beneficial for the reader. It is a plus that the library support some LLMs.

+ The paper is well presented and easy to read.

**Weaknesses:**

- The paper does not include new techniques or advancements in the state of the art. It entirely relies on attacks and defenses that have already been proposed in the research literature. The experimental evaluation is a confirmation that the code produces reasonable results, but there is no novelty in there either.

- The authors did not provide the code implementation of the library (I believe it could be easily anonymized). Then, as the main contribution of the paper is the library, not having access to the code implementation makes it hard to assess its characteristics and possible weaknesses.

- The coverage of the library misses attacks at test time (e.g., adversarial examples) and other privacy attacks that can happen both during the training and the deployment of federated learning models, such as property inference attacks or membership inference attacks, to cite some.

**Questions:**

+ Could the authors provide the whole catalogue of attacks and defenses implemented in the library?

+ Could the authors provide the code implementation (anonymized).

+ Could the authors clarify the contributions of the paper for advancing the state of the art in robust federated learning?

---

> ### Author Response · Authors · 2023-11-21
> **Response to Reviewer D1fe**
>
> We appreciate your feedback and your effort in reviewing our paper. Below are our responses.
>
> **W1 & Q3: Limited novelty**
>
> Response: As stated on the ICLR website (https://iclr.cc/Conferences/2024/CallForPapers), the conference invites papers that are in the area of datasets and benchmarks. Our paper directly responds to the ICLR's emphasis on datasets and benchmarks. Our benchmark is designed to implement the state-of-the-art attack and defense mechanisms to provide a comprehensive and rigorous testing ground for these techniques. With this comprehensive benchmark, we believe the community can further advance the research on secure federated learning.
>
> **W2 & Q2: Code implementation**
>
> Response: The reason why we did not include our code implementation was because FedSecurity is integrated into an open-sourced Federated Learning (FL) library, and anonymizing such integrated code without losing its essence and functionality is significantly challenging. Acknowledging your concern about the difficulty in thoroughly assessing the library, we have now extracted the security-related components from the open-sourced library, which preserves the integrity and functionality of the benchmark while ensuring anonymity; see [1]. Specifically, we implemented 8 attacks and 14 defenses that are widely considered in literature; see [2] for a full list of attacks and defenses we implemented. We hope this addition addresses your concerns and aids in the thorough assessment of our work.
>
> **W3. Cite some privacy attacks**
>
> Thanks for your suggestions! We have cited the following privacy attacks [5,6,7,8,9]. The reason why we did not include a number of privacy attacks in the benchmark is that privacy attacks usually require strong assumptions, e.g., in DLG attack [3], to revert an image training data that can be recognized by the naked eye, the model should be trained using only one sample, and the model inversion attack [4] requires knowing the distribution of the training data in advance. In real-world applications, it’s hard to apply such attacks. So, we decided to focus more on the attacks that are more practical in real-world scenarios.
>
> **Q1: List of implemented attacks and defenses**
>
> Please refer to [2] for the list of implemented attacks and defenses. We also listed them in the introduction. The description in the paper is as follows:
> ```
> ​​FedSecurity implements attacks that are widely considered in the literature, including Byzantine attacks of random/zero/flipping modes (Chen et al., 2017; Fang et al., 2020), label flipping backdoor attack (Tolpegin et al., 2020), deep leakage gradient (Zhu et al., 2019), and model replacement backdoor attack (Bagdasaryan et al., 2020). Some of the well-known defense mechanisms supported include Norm Clipping (Sun et al., 2019), Robust Learning Rate (Ozdayi et al., 2021), Krum (and m-Krum) (Blanchard et al., 2017), SLSGD (Xie et al., 2020), geometric median (Chen et al., 2017), weak DP (Sun et al., 2019), CClip (Karimireddy et al., 2020), coordinate-wise median (Yin et al., 2018), RFA (Pillutla et al., 2022), Foolsgold (Fung et al., 2020), CRFL (Xie et al., 2021), and coordinate-wise trimmed mean (Yin et al., 2018).
> ```
>
>
>
> [1] Code implementation: https://gitfront.io/r/Submission1595/4rUJj4M2NPzy/FedSecurity/
>
> [2] List of implemented attack and defense mechanisms: https://gitfront.io/r/Submission1595/4rUJj4M2NPzy/FedSecurity/blob/readme.md
>
> [3] Zhu, Ligeng, Zhijian Liu, and Song Han. "Deep leakage from gradients." Advances in neural information processing systems 32 (2019).
>
> [4] Geiping, Jonas, et al. "Inverting gradients-how easy is it to break privacy in federated learning?."
> Advances in Neural Information Processing Systems 33 (2020): 16937-16947.
>
> [5] Fowl, Liam, et al. "Robbing the fed: Directly obtaining private data in federated learning with modified models." arXiv preprint arXiv:2110.13057 (2021).
>
> [6] Wang, Zhibo, et al. "Poisoning-assisted property inference attack against federated learning." IEEE Transactions on Dependable and Secure Computing (2022).
>
> [7] Luo, Xinjian, et al. "Feature inference attack on model predictions in vertical federated learning." 2021 IEEE 37th International Conference on Data Engineering (ICDE). IEEE, 2021.
>
> [8] Zhang, Jingwen, et al. "Gan enhanced membership inference: A passive local attack in federated learning." ICC 2020-2020 IEEE International Conference on Communications (ICC). IEEE, 2020.
>
> [9] Melis, Luca et al. “Exploiting Unintended Feature Leakage in Collaborative Learning.” 2019 IEEE Symposium on Security and Privacy (SP) (2018): 691-706.

---

### Meta-Review · Area_Chair_tpEX · 2023-12-06

**Metareview:**

This paper presents FedSecurity, a benchmark designed for simulating attacks and defenses in federated learning (FL). FedSecurity comprises two key components: FedAttacker, which injects attacks such as data poisoning, model poisoning, and data reconstruction, and FedDefender, which implements defenses like clipping, robust aggregation, and adding noise. The benchmark allows customization of attacks and defenses through provided APIs and flexibility in configuring various models, datasets, and FL optimizers. Experiment results demonstrate that Byzantine attacks, such as random noise, can significantly degrade accuracy, while defenses like m-Krum can mitigate these effects. However, defenses may unintentionally harm accuracy even when no attack occurs.

There are notable limitations in the benchmark:

Limited Defense Mechanisms: Only a small subset of defenses from the literature is currently implemented.

Limited Analysis: More in-depth analysis and visualization illustrating how attacks and defenses impact model convergence would provide valuable insights.

Outdated Baselines: The implemented attack/defense methods were published before 2021. The authors are encouraged to reproduce more SOTA attacks/defenses to improve the utility of this benchmark further.

Small LLM Experiments: Evaluations on large language models are restricted to BERT and Pythia. Including experiments with a more diverse set of LLMs would bolster this aspect.

After rebuttal, reviewers commented that the experiments conducted can be improved and evaluation on scenarios with more clients to test the scalability is necessary, and the contribution of this paper is a bit limited. Considering these limitations, I recommend rejection.

**Justification For Why Not Higher Score:**

N/A

**Justification For Why Not Lower Score:**

N/A

---

### Decision · Program_Chairs · 2024-01-16

Reject